# Full-frame and high-contrast smart windows from halide-exchanged perovskites

You Liu [1,4], Jungan Wang[1,4], Fangfang Wang[1,4], Zhengchun Cheng[1], Yinyu Fang[1], Qing Chang[1], Jixin Zhu [1], Lin Wang [1], Jianpu Wang [1], Wei Huang [2,3✉] & Tianshi Qin [1✉]

Window glazing plays an essential role to modulate indoor light and heat transmission, which is a prospect to save the energy cost in buildings. The latest photovoltachromic technology has been regarded as one of the most ideal solutions, however, to achieve full-frame size (100% active area) and high-contrast ratio (>30% variable in visible wavelength) for smart window applicability is still a challenge. Here we report a photovoltachromic device combining full-transparent perovskite photovoltaic and ion-gel based electrochromic components in a vertical tandem architecture without any intermediated electrode. Most importantly, by accurately adjusting the halide-exchanging period, this photovoltachromic module can realize a high pristine transmittance up to 76%. Moreover, it possesses excellent colour-rendering index to 96, wide contrast ratio (>30%) on average visible transmittance (400-780 nm), and a self-adaptable transmittance adjustment and control indoor brightness and temperature automatically depending on different solar irradiances.

[1] Key Laboratory of Flexible Electronics (KLOFE) & Institution of Advanced Materials (IAM), Jiangsu National Synergetic Innovation Center for Advanced Materials (SICAM), Nanjing Tech University (NanjingTech), Nanjing, Jiangsu, China. [2] Key Laboratory for Organic Electronics & Information Displays (KLOEID) & Institute of Advanced Materials (IAM), Nanjing University of Posts and Telecommunications, Nanjing, Jiangsu, China. [3] Frontiers Science Center for Flexible Electronics & Institute of Flexible Electronics (IFE), Northwestern Polytechnical University (NPU), Xi'an, Shaanxi, China. [4] These authors contributed equally: You Liu, Jungan Wang, Fangfang Wang. ✉email: iamwhuang@nwpu.edu.cn; iamtsqin@njtech.edu.cn

Since the original photoelectrochromic devices (PECDs) based on liquid dye-sensitized solar cells (DSSCs) was described by Bechinger et al.[1], the development on self-driven smart windows has been evolving for quarter century[2]. One step forward for PECDs was disclosed by Wu et al.[3], who presented the first photovoltachromic devices (PVCDs) combining the photovoltaic (PV) and electrochromic (EC) features of DSSCs in a separated electrode architecture[3]. This breakthrough broadened the PV technologies to all solid-state candidates, such as amorphous silicon (α-Si), organic photovoltaic (OPV)[4], and perovskite solar cell (PSC)[5]. Nevertheless, the integration of solid PVCDs for building envelopes was filled with more challenges than liquid PECDs. Starting from the external connections between separated opaque PV and EC devices, these systems were not truly monolithic structure devices, and in simple terms, merely shortened power delivery distance from the photovoltaic power generator to terminal electrical equipment, which were too cumbersome for building integrated applications. Soon afterwards monolithic integration was succeeded by a parallel side-by-side architecture. Unfortunately, the opaque or semi-transparent PV component was unable to be chromotropic as an inactive area, limited its application as ideal smart windows requiring 100% active coverage (Fig. 1a). Ultimately, vertical tandem architecture was considered as the most advanced integrating strategy. This monolithically hierarchy configuration demanded a high visible transparency for both PV and electrode layers. Otherwise these layers would block the visible sunlight through smart windows for indoor illumination. In view of this, recently semi-transparent characteristics had applied into tandem structure PVCD by using two pathways: (i) Physically reducing the thickness or patterning voided windows on intrinsically opaque PV layers led to low pristine transmittance (T%) and low contrast ratio in PVCDs (Supplementary Table 1). (ii) Chemically broadening the optical bandgap of PV materials to absorb only partial visible wavelength resulted in an uncomfortable visual environment and poor color-rendering index (CRI) (PVCDs appeared orange or brownish, since they only absorbed blue-green radiation). In this work, we present a facile full-solution processing technique to realize a full-transparent perovskite PV layer with high pristine T% and excellent CRI. Based on this we manufacturing a monolithic photovoltachromic smart window using visibly-transparent perovskite and flexible electrochromic ion-gel[6].

## Results

**High quality MAPbCl₃ films based on ion-exchanged reaction.** Metal halide perovskites become state-of-the-art in the optoelectronic field because of their high power conversion efficiency (PCE), abundant diversified ingredient, and tunable absorption across the whole visible spectrum[7,8]. Transparent photovoltaic technology provides an effective way to fabricate power generation smart windows, building integrated photovoltaics, agricultural greenhouses, and other fields[9]. However, compared to iodide and bromide counterparts, chloride perovskite has been rarely exploited in PV application due to its limited solution-processability and over-grown crystallinity, leading to poor film morphology and massive grain boundary recombination. Thus the photograph of Methylamine lead chloride (MAPbCl₃) perovskite film prepared by one-step solution (named 1-step MAPbCl₃) spin coating method exhibits a frosted appearance with high haze (Fig. 1b). Herein, we have developed a halide-exchanging technology (Supplementary Fig. 1), which could convert the solution-processable bromide perovskite (MAPbBr₃) film with into full transparent perovskite (named 2-step MAPbCl₃) with a low haze for further enhancing film transmittance (Fig. 1b). Semi-transparent MAPbBr₃ perovskite

layer was originally composed of condensed grains, then shrunk along grain boundaries during halide-exchanging process, and finally formed transparent 2-step MAPbCl₃ film in 15 min (Fig. 1b). By contrast, 1-step MAPbCl₃ films have the morphology of isolated cubic blocks with the scale of tens of microns. The T% spectra of perovskite films (Fig. 1c) exhibited that the shortest transparent cut-off wavelength was shifted from 525 nm of pristine Methylamine lead bromide (MAPbBr₃) to 407 nm of 15-min ion exchange 2-step MAPbCl₃ and the average visible transmittance (AVT) (400–780 nm) was raising along with the halide-exchanging period from pristine 44–76% for 15 min. It was worth noting that although the absorption cutoff wavelength of the 1-step MAPbCl₃ film was also around 405 nm, the transmissivity of visible light was very poor with a AVT of 15%. It was obvious that the result is directly related to the haze of the film. In order to prove this point, the haze values of three perovskite films were characterized, manifesting 95.8% of 1-step MAPbCl₃, 32.2% of MAPbBr₃, and 34.6% of 2-step MAPbCl₃ perovskite films, respectively. The X-ray diffraction spectrum (XRD) was performed to confirm the incorporation of Cl into the MAPbBr₃ lattice (Fig. 1d). The diffraction peaks at 15.0° and 30.2° from the pristine sample were ascribed to the (100) and (200) of MAPbBr₃ crystal, respectively[10]. During halide anion exchange, these characteristic peaks moved toward higher angles because of crystal lattice shrinkage. After 15 min ions exchange peaks shifted to 15.6° and 31.4°, respectively, which were coincided with the characteristic diffraction peaks of the cubic phase of MAPbCl₃ as previously reported[11], suggesting the completed conversion of MAPbBr₃ into MAPbCl₃. In meanwhile, there are diffraction peaks at 22.2° and 35.3° of 1-step MAPbCl₃ film attributing to (110) and (120), and extra signals from FTO substrate, which indicates that the 1-step MAPbCl₃ film has poor growth orientation and many covering defects on substrate.

**Optimization of MAPbCl₃ perovskite film for PVCD.** In order to find out the optimal halide ingredient in perovskite for PVCD, the halide-exchanging progress was further investigated. The extrapolation of the linear part of the Tauc plot (Fig. 2a) showed the optical gap of MAPbBr$_x$Cl$_{3-x}$ were shifted from 2.24 eV to 2.95 eV, corresponding to band gaps of pristine MAPbBr₃ and MAPbCl₃ perovskites[12,13]. It was noticed that, after 10-minute anion exchange, the band gap shifted to 2.89 eV, enabling saturated CRI and AVT. The synchrotron-based 2D grazing-incident wide-angle X-ray scattering (GIWAXS) was conducted on a series of samples, including pristine MAPbBr₃ and MAPbBr$_x$Cl$_{3-x}$ perovskites with different halide-exchanging periods (Fig. 2b). The coordinate of in-plane direction ($q_{xy}$) and out-of-plane direction ($q_z$) diffractions corresponding to perovskite (110) lattice planes are obviously shifted from (1.07, 1.08) for MAPbBr₃ to (1.12, 1.12) for MAPbBr$_x$Cl$_{3-x}$ (10 min) because of crystal lattice shrinkages (Supplementary Fig. 2). To investigate the influence of the anion exchange on the HOMO energy level of MAPbBr$_x$Cl$_{3-x}$, the HOMO energy level of perovskites with different reaction periods were determined by photoelectron yield spectroscopy (PYS). Where a molecular of 2,2′,7,7′-tetrakis[N,N-phenothiazine]-9,9′-spirobi-fluorene (Spiro-PT)[14] was employed as hole transport material (HTM) because of matchable HOMO energy level (Fig. 2c). The detailed elemental distribution in PV component was further analyzed by time-of-flight secondary-ion mass spectra (ToF-SIMS) (Fig. 2d), which reveals clear boundaries of functional layers ascribed to anion counts of S⁻, Br⁻, Cl⁻, and F⁻. Additionally, doping information could be observed more clearly in the corresponding 3D distribution images (Fig. 2e). Notably, the intensity of Cl⁻ is increasing whereas that of Br⁻ is decreasing from bottom to top side, in accordance with the halide-diffusion mechanism from solution to film. Therefore, based on the halide exchanged strategy,

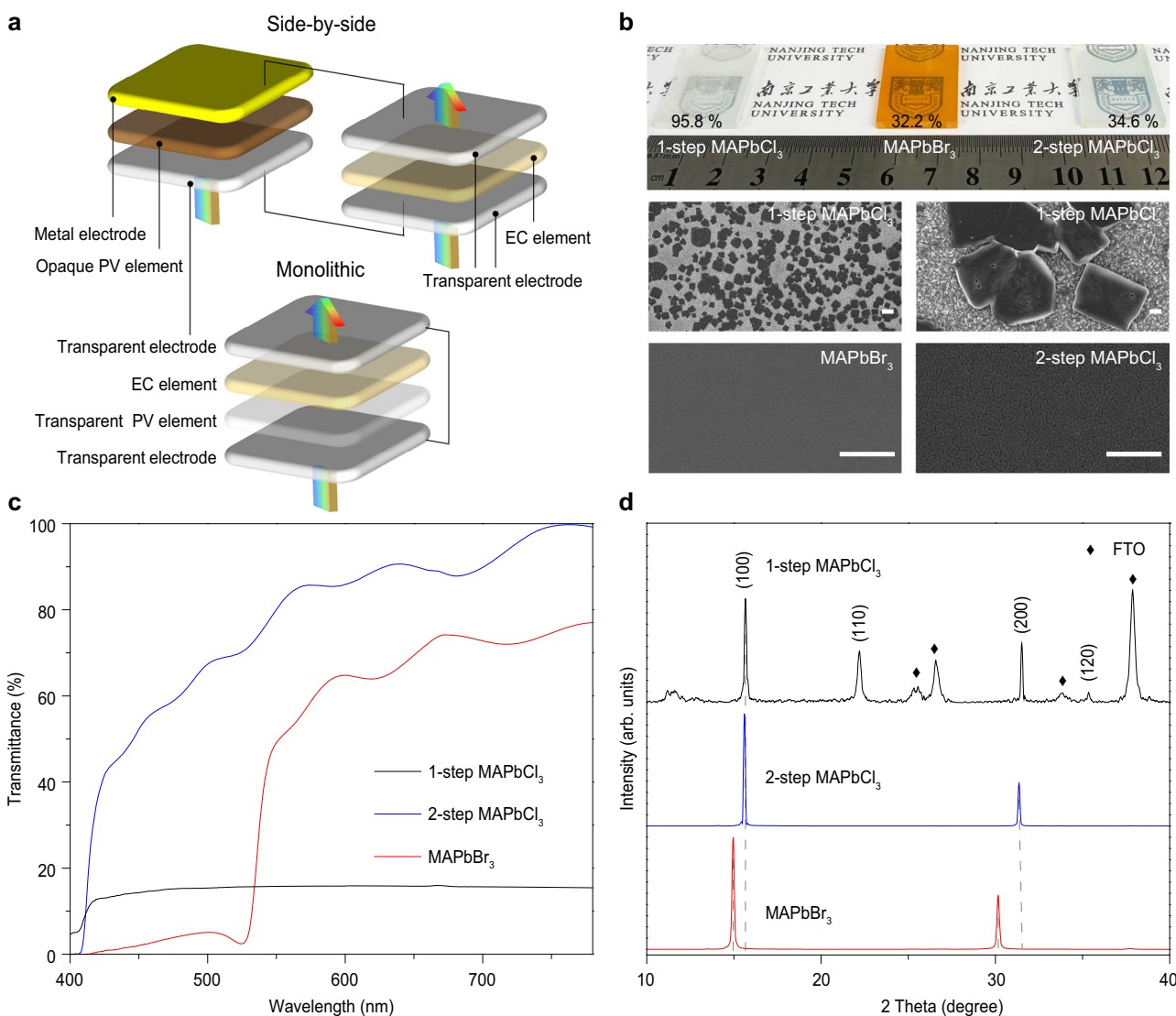

**Fig. 1 Evolution on architectures of PVCDs and characterization of perovskite films. a** Four-terminal side-by-side PV-EC device architecture: two terminals of PV element and two terminals of EC element were connected via two external circles. Two-terminal monolithic integrated PVCD architecture in this work: only one external circle required to connect top and bottom terminals, which is the most ideal structure for achieving full-frame active area and high transmittance synchronously. **b** The photographs and SEM images of 1-step MAPbCl$_3$, MAPbBr$_3$ and 2-step MAPbCl$_3$ film respectively. Scale bar is 5 μm. **c** The transmittance spectra of 1-step MAPbCl$_3$, MAPbBr$_3$ and 2-step MAPbCl$_3$ films. **d** X-ray diffraction patterns of 1-step MAPbCl$_3$, MAPbBr$_3$, and 2-step MAPbCl$_3$ films.

the Cl/Br ratio in perovskite is a gradient distribution in vertical direction, resulting in a graded heterojunction with variable molecular orbital energy levels[15], whose functionality in photoelectronic devices will be discussed afterwards.

Solar-driven smart window plays an essential role to modulate indoor illumination and temperature, which is the most ideal glazing technology to save the energy cost in buildings. To date, many prototypes of PVCDs based on liquid dye-sensitized solar cells or side-by-side device structure or opaque perovskite layers or four-terminal external circuits have been reported (Supplementary Table 2)[2,16–18]. It is still a challenge to realize the ideal solar-driven smart window in solid state with simplest two-terminal electrodes in full-frame size. In order to achieve such ultimate smart window architecture, we fabricated a monolithic PVCD as glass/FTO/ PV/electrochromic (EC) gel/FTO/glass architecture via a full solution process (Supplementary Fig. 3), which presented a feasibility for scale-up large-area device (ca. 36 cm$^2$) (Supplementary Fig. 4). This sandwiched PVCD architecture

was unambiguously observed by cross-section SEM images (Fig. 2f) in which the flexible EC gel[6] (~90 μm in thickness) exhibited sufficient ohmic contact and mechanical strength on the top of the PV component (~800 nm in thickness). The partially enlarged details presented the PV submodule consisted of 300 nm ETL, 350 nm perovskite and 40 nm HTL.

To further understand the working mechanism that occurred in this monolithic PVCD, we separately prepared individual PV and EC components and analysed their energy levels. The valence bands (VBs) and conduction bands (CBs) of Spiro-PT and perovskites with different reaction-diffusion periods were calculated by PYS (Fig. 2a) and cut-off edge of absorption spectra (Fig. 2c), which are in accordance with literatures[19]. The energy level diagram of PVCD was shown in Fig. 2g, in which the Shockley–Queisser (SQ) theory limit of PV component ($\Delta E_{PV}$) is 1.6 eV and EC component ($\Delta E_{EC}$) is 1.0 eV, respectively. It is well known that graded heterojunctions could improve the photo-charge collection and reduce recombination loss[15,20–23]. In the

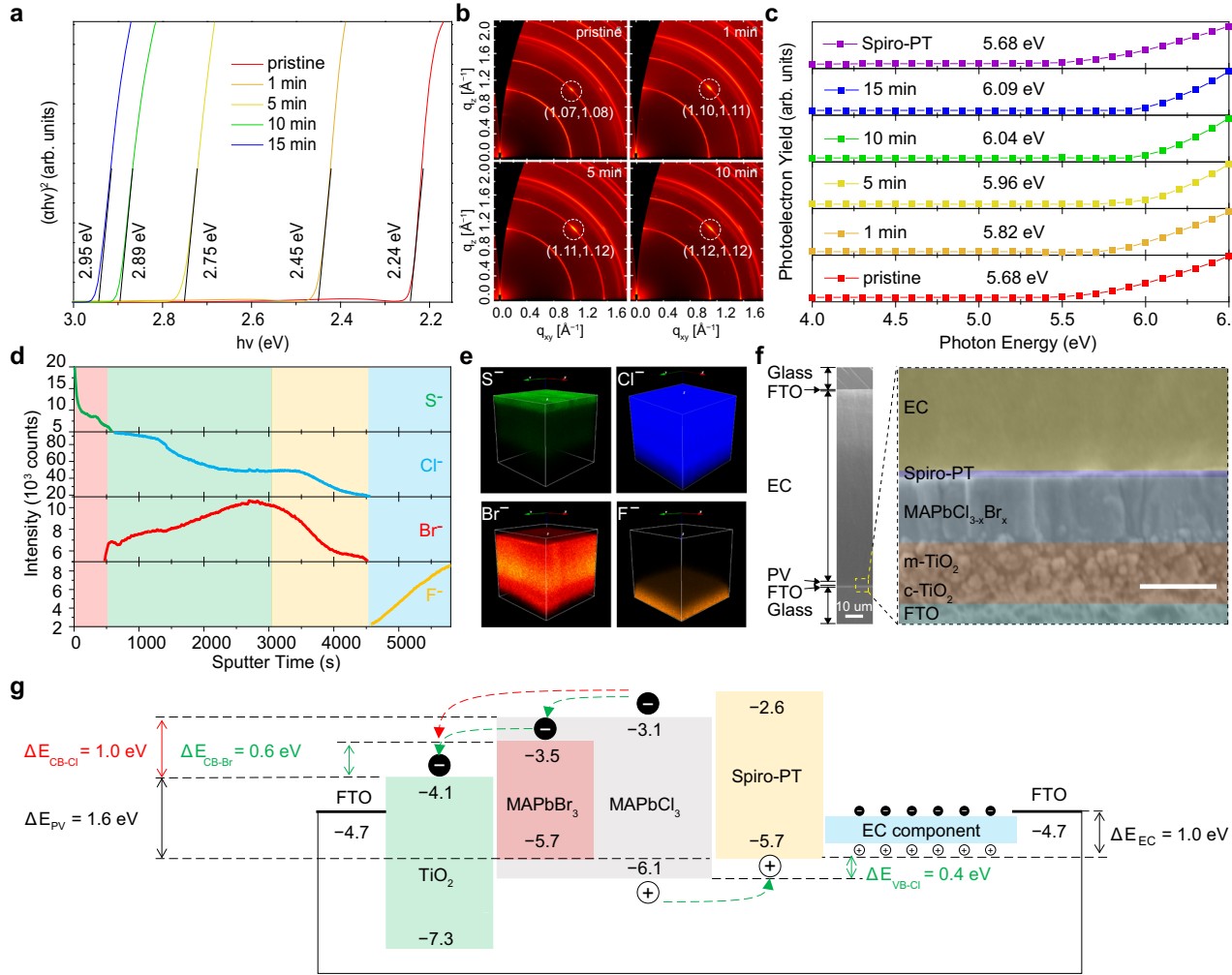

**Fig. 2 Characterization analysis of PV components from semi-transparent to full-transparent and work mechanisms of monolithic PVCD. a** Tauc plots of perovskites treated with halide diffusion periods of 0 min, 1 min, 5 min, 10 min and 15 min, respectively. **b** 2D-GIWAXS patterns of pristine $MAPbBr_3$ and $MAPbBr_xCl_{3-x}$ perovskites with a halide-exchanging period for 1, 5, and 10 min. **c** Valence band levels of Spiro-PT and $MAPbBr_xCl_{3-x}$ treated with different halide-diffusion periods. The VB values were determined by PYS. **d** ToF-SIMS depth profiles showed the concentration of selected species across the PV semi-module based on halide-exchanged $MAPbBr_xCl_{3-x}$ perovskite (10 min). **e** Reconstructed elemental 3D maps of negative ions traced in the depth profile. The $xy$ dimensions of the analyzed area of the films were 80 μm × 80 μm. **f** Cross-section SEM images of a monolithic PVCD as glass/FTO/PV/ EC/FTO/glass device structure, with the detailed architecture of PV semi-module as $c$-$TiO_2$/$m$-$TiO_2$/$MAPbBr_xCl_{3-x}$/Spiro-PT. Scale bar is 500 nm. **g** Schematic energy levels of monolithic PVCD, in which Br-rich perovskite in halide-diffused perovskites is one key factor. It can suppress the non-radiative recombination of large $\Delta E_{LUMO}$ and improve electron extraction from perovskite to $TiO_2$ due to its intermediate LUMO level between $\Delta E_1$ and $\Delta E_2$.

PV component, both Cl-rich and Br-rich perovskites possess matchable VBs ($\Delta E_{VB-Cl} = 0.4$ eV and $\Delta E_{VB-Br} = 0$ eV) to HTL, inducing a fine hole-extraction for positive charge accumulation to drive EC. However, on the other side, Cl-rich perovskite shows a considerable energy gap of LUMO ($\Delta E_{CB-Cl} = 1.0$ eV) to ETL, whereas Br-rich perovskite is more matchable ($\Delta E_{CB-Br} = 0.6$ eV) to smoothly extract photo-electrons into ETL and further transport to the other side FTO electrode of EC via the external circuit. This might be the reason that independent halide-diffused perovskite PVs demonstrated a reduction on both open-circuit voltage ($V_{OC}$) and short-circuit current density ($J_{SC}$) along with elongating ion-exchanged periods (Supplementary Fig. 5 and Supplementary Table 3). Therefore, one key factor to realize a working monolithic PVCD is keeping Br-rich perovskite residually at the ETL side. Fortunately, our aforementioned ToF-SIMS analysis (Fig. 2d) presented that the reaction-diffusion process could create a gradient Cl/Br distribution in perovskite, which offered an intermediate energy ladder of Br-rich perovskite in between Cl-rich perovskite and ETL for better interfacial

charge collection. In addition, incident photon-to-electron conversion efficiency (IPCE) spectra (Supplementary Fig. 6) can explain that partial root of descending $J_{SC}$ is due to narrowing light conversion wavelength range.

**High reversibility and stability of PVCD.** By optimizing the halide-diffusion period for perovskite, we successfully achieved a self-adaptable adjustment on transmittances depending on varying solar irradiation intensities for this monolithic PVCD. We performed PVCD based on 10-min halide-diffused perovskite under varying solar intensity from 10 to 100% AM 1.5G illumination (0.1–1.0 sun) (Fig. 3a). The initial T% is around 83% at 600 nm for the bleached state (BS) of PVCD. It is obvious that under different sunlight irradiations, the steady T% for the colored state (CS) is divergent from 83% (0.1 sun) to 20% (1.0 sun). In contrast, PVCD based on semi-transparent perovskite demonstrates a much narrower contrast difference between 0.1 and 1.0 sun (Supplementary Fig. 7) due to its visible-wavelength photon-to-electron conversion providing saturated

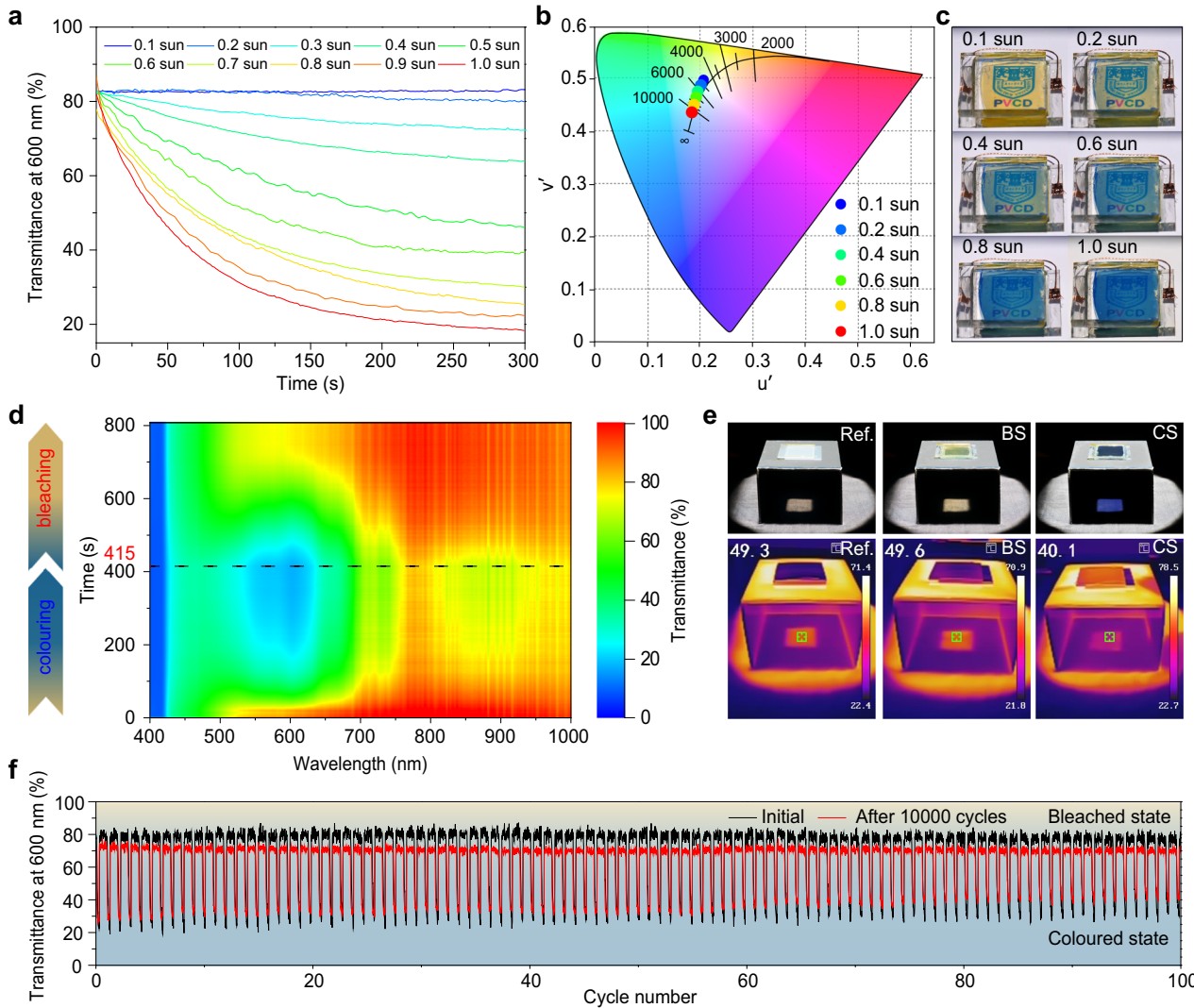

**Fig. 3 Photoelectronic performances of solar-adaptable monolithic PVCD based on full-transparent PV. a** Real-time transmission ($T\%$-t) tendencies at 600 nm under different sunlight intensities from 0.1 to 1.0 sun. **b** CIE 1976 ($u'$, $v'$) coordinates and **c** photographs of PVCD at different sunlight intensities. **d** Contour mapping diagram of real-time full-wavelength (400–1000 nm) transmission spectra under AM 1.5G illumination. External wires of PVCD were connected as close circuit from 0 to 415 s for coloring process, and disconnected as open circuit from 415 to 800 s for bleaching process. **e** Profile and IR photos of PVCD at BS and CS, two FTO glass substrates used as reference. **f** Repeatability of PVCD switching between BS and CS by using 395 nm UV-LED as control. In each cycle, PVCD was illuminated for 10 s for coloring process and blocked for 45 s for bleaching process, periodically. Black line records the initial 100 cycles, after 1-week continuous service period (>10,000 cycles) by directly connecting external circuit of PVCD in glove box, red line records the final 100 cycles.

current density. Moreover, a photochromic lens as reference also manifests a poor contrast ratio (Supplementary Fig. 7), since the trigger of photochromic compounds is independent of sunlight intensity[24]. Besides that, according to the CIE 1976 color space standard, color coordinates of the transmission through PVCD demonstrated good color-neutralities close to Planckian locus for all diverse sunlight intensities, only with transitional correlated color temperatures (CCT) from 4800 to 10,400 K (Fig. 3b). Photographs of PVCD under 0.1 sun illumination clearly showed an excellent color rendering index (CRI) of 96 (Fig. 3c), which emerged "excellent level" color saturations of indoor objects by using this PVCD-based smart window, according to the international standard—ISO 9050 glass in building[25].

The in situ dynamic $T\%$ spectra of visible and near-infrared (IR) wavelengths were investigated under 1.0 sun illumination as the contour mapping diagram (Fig. 3d), wherein the external circuit between two FTO electrodes was disconnected at the 415th

second. The instantaneous $T\%$ spectra of PVCD present the photovoltachromic coloring period is ca. 200 s and self-bleaching time by disconnection is ca. 300 s (Supplementary Fig. 8), exhibiting a perfect contrast ratio (>30%) on average visible-light transmittance (AVT) between BS and CS. To mimic the effect of PVCD on indoor dimming and cooling, we also collected profile and IR photos for three different situations (Fig. 3e). In contrast to the reference temperature by two glass substrates at 49 °C, the PVCD demonstrates no interference on temperature at 49 °C for BS and a significant cooling-down to 40 °C for CS, heralding an enormous energy-saving on air-conditions for buildings. To estimate the duration of transmission response under operational conditions, the AVT of PVCD was monitored by repeatedly switch on-off 395 nm UV controlling light (Supplementary Fig. 9), which demonstrated an good >10,000 cycle reproducibility and one-week service period (Fig. 3f). Although the contrast ratio of ~60% at 600 nm in the first 100 cycles decreased to ~40% after

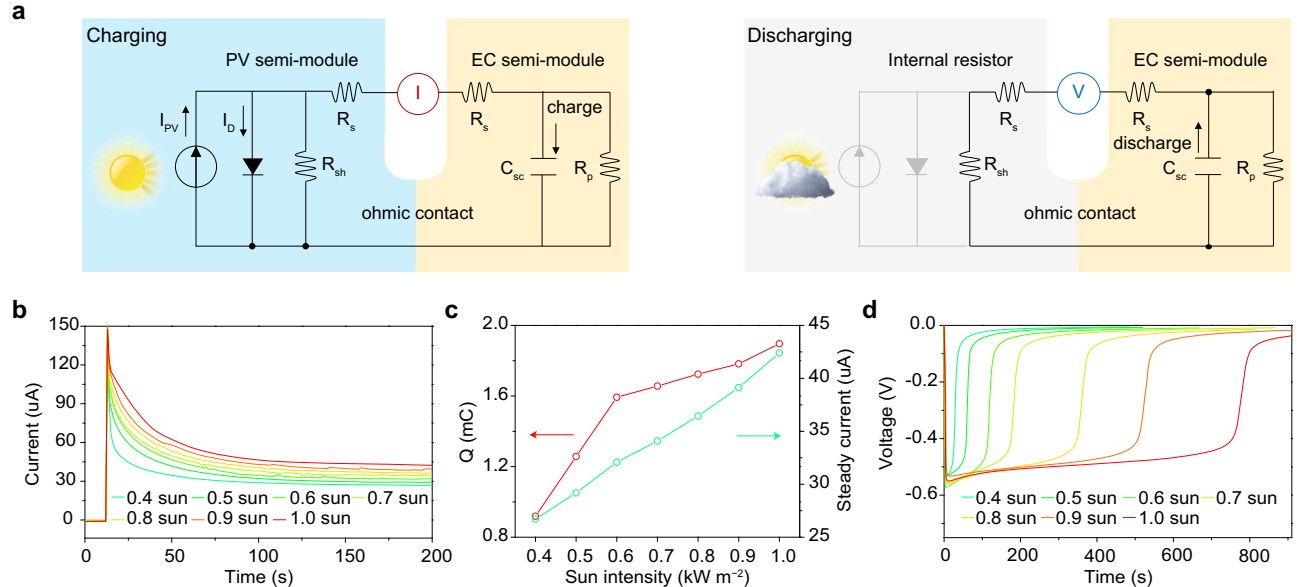

**Fig. 4 Working principles of solar-adaptable PVCD. a** Charging state and discharging state circuit diagrams of solar-adaptable PVCD. **b** Real-time current ($I-t$) tendencies in external circuit of PVCD during charging process under different sunlight intensities. **c** The capacitance ($Q$) of EC components after the charging process is completed for PVCD charged by different sunlight intensities. **d** Real-time voltage ($V-t$) tendencies in external circuit during discharging process for PVCD charged by different sunlight intensities.

10,000 cycles, this stability and reproducibility are still considerable compared with the previously reported performances of PVCDs (Supplementary Table 2).

**The self-adaptability of PVCD.** Based on our postulated electric circuit diagrams for charging and discharging states (Fig. 4a), to further understand the working mechanism that occurred in our PVCD, we in-situ monitor short-circuit current ($I_{SC}$) for charging state at different solar irradiances and corresponding open-circuit voltage ($V_{OC}$) for discharging state, respectively. The $I-t$ diagram (Fig. 4b) during the charging procedure demonstrates a two-step tendency including an exponential decay (from $I_{max}$ to $I_{steady}$) at the beginning and a steady current ($I_{steady}$) afterwards, which is corresponding to charge capacitor $C_{SC}$ and pass resistor $R_P$, respectively. Based on the following formulas:

$$Q_{SC} = \int_{max}^{steady} I(t)dt \qquad (1)$$

Both calculated short-circuit quantity of charge ($Q_{SC}$) and experimental $I_{steady}$ are increasing along with brightening sunlight (Fig. 4c), which can explain the working principle of solar-adaptable adjustment on T% and prove the equivalent circuit of monolithic PVCD. When sunlight irradiates PVCD, the PV semi-module will provide voltage to charge the EC semi-module. The $Q_{SC}$ of PVCD is in direct proportion to T%, which can be adjusted by the photon-generated current during exponential decay step. Once EC semi-module charged to $Q_{SC}$, $R_P$ will stabilize $I_{steady}$ and bypass photon-generated current into EC for further darkening. Besides that, when we block the sunlight or disconnect the external circuit between two electrodes, the electrochemical capacitor will self-discharge via the drain current circuit through $R_P$, resulting in the self-bleaching of PVCD. The instantaneous maximum discharging voltage ($V_{EC}$) in $V-t$ diagram (Fig. 4d) is steady at ca. 0.6 V for different sunlight intensities and consistent with the driving voltage of EC ion-gel (Supplementary Fig. 10), which is in accord with the fundamental mechanism of electrochemical capacitors[26].

## Discussion

In summary, we initially applied self-regulated anion exchange strategy in photoelectronic application, and successfully realized a promising enhancement performances, synchronously. This full-transparent monolithic smart window possesses not only full advantages of self-power, on-demand control and self-adaptive based on solar-irradiances, but also good features including excellent visible transparency and color neutrality, high adjustable range on transmittance and temperature, as well as repeatable serviceability. Moreover, the level for switching AVT could be self-adaptable by solar irradiation intensity automatically to endue smart windows a "smarter brain" to response sunrise-sunset and real weathers (Supplementary Movie 1). Most importantly, the full-solution processability of two-terminal sandwiched glass-gel-glass architecture can offer the simplest architecture, scalable producibility and building integration as safety glazing for future energy-saving skyscrapers.

## Methods

**Materials.** Methylamium bromide (MABr, 99.99%), methylamium chloride (MACl, 99.99%), and lead bromide (PbBr$_2$, 99.99%) were purchased from Hang-zhou Perovs Optoelectronic Technology Corp (China). 4-tert-butylpyridine (TBP, 96.0%), heptyl viologen dibromide (HV(Br)$_2$, 97%), 1,1′-dimethylferrocene (dmFc, 95%), 1-Butyl-3-methylimidazolium bis(trifluoromethylsulfonyl)imide ([BMI] [TFSI], ≥98.0%), and poly(vinylidene fluoride-co-hexafluoropropylene) (PVDF-HFP) were purchased from Sigma-Aldrich. Lithium bis(trifluoromethanesulfonyl) imide (Li-TFSI, 98+%), and solvents including N,N-dimethylformamide (DMF, 99.8%), dimethyl sulfoxide (DMSO, 99.9%), isopropanol (IPA, 99.5%), chlorobenzene (CB, 99.9%), ethyl acetate (EA, 99.5%), and acetonitrile (99.5%) were purchased from Alfa Aesar. All chemicals and solvents were used without further purification.

**Solutions for PV component.** Compact TiO$_2$ (c-TiO$_2$) solution was prepared by adding dropwise HCl solution (35 μL of 2 M HCl in in 2.53 mL ethanol) into titanium isopropoxide (370 μL in 2.53 mL ethanol). The mixture was filtered with a PTFE 0.22 μm filter (SIMADZU-GL) before spin-coating. Mesoporous TiO$_2$ (m-TiO$_2$) solution was prepared by diluting Dyesol 18NR-T paste into α-terpineol and ethanol in wt% of m-TiO$_2$:α-terpineol:ethanol = 2:6:3. MAPbBr$_3$ perovskite precursor solution was prepared by dissolving MABr (1.2 M) and PbBr$_2$ (1.2 M) in a mixed solvent of DMF/DMSO (7/3, v/v). The prepared solution was stirred at room temperature over night before use. MACl solution was prepared by dissolving in iso-proponal solvent with a concentration of 10 mg mL$^{-1}$. Spiro-PT solution was prepared by dissolving in chlorobenzene solvent with a concentration of 15 mg mL$^{-1}$.

**Gels for EC component**. Heptyl viologen bis(trifluoromethylsulfonyl)imide (HV (TF-SI)$_2$) was prepared by an ion-exchange reaction between 514 mg HV(Br)$_2$ (1 eq) and 689 mg Li-TFSI (2.4 eq) in 40 mL deionization water. The aqueous solution was kept shaking for 3 h, then centrifuged at 3000 rpm for 5 min. The precipitated product HV(TFSI)$_2$ was washed by deionization water until color from yellow to white, then dried by cryodesiccation. Twenty-one milligram HV(TFSI)$_2$ as aforementioned and 7 mg dmFc was dissolved in 420 mg [BMI][TFSI] at 60 °C for 30 min, then added dropwise under stirring into another solution which was prepared by 84 mg PVDF-HFP in 500 mL EA, resulting in the EC gel.

**Transparent PV component fabrication**. The pre-patterned FTO (sheet resistance: 8 Ω sq$^{-1}$, thickness: 2.5 mm, Asahi Glass Co.) substrates (2 cm × 2.5 cm) were ultrasonically cleaned with diluted detergent, deionized water, acetone, and iso-propanol in succession for 20 min. The as-cleaned FTO substrates were treated by UV-ozone for 30 min. The c-TiO$_2$ solution was spin-coated onto the FTO substrates at 2000 rpm for 20 s with a ramping rate of 2000 rpm s$^{-1}$, and then the samples were heated at 180 °C for 10 min. After cooling to room temperature, the m-TiO$_2$ paste was spin-coated at 2000 rpm for 20 s with a ramping rate of 2000 rpm s$^{-1}$, then annealed at 120 °C for 10 min. These ETL-coated substrates were sintered at 500 °C for 30 min in a muffle furnace. After cooling to room temperature, these ETL coated substrates were transferred into a N$_2$-filled glovebox. Hundred microliter of MAPbBr$_3$ precursor solution was spin-coated by a two-consecutive step program at 1000 rpm for 10 s with a ramping rate of 1000 rpm s$^{-1}$, and 4000 rpm for 60 s with a ramping rate of 4000 rpm s$^{-1}$. During the second step, 100 μL of chlorobenzene was drop on the center of the spinning substrates at 40 s prior to the end of the whole spinning program. These MAPbBr$_3$ perovskite films were then annealed at 100 °C for 10 min in glovebox. For halide diffusion process, the as-prepared MAPbBr$_3$ perovskite film was immersed into 5 mL of as-prepared MACl solution at room temperature in glovebox for different periods (1–30 min), then rinsed with iso-propanol three times and dried at 70 °C for 10 min in glovebox, indicating per-ovskites with different transmittances. Afterward, 40 μL of as-prepared Spiro-PT solution was spin-coated onto the perovskite films at 4000 rpm for 30 s with a ramping rate of 4000 rpm s$^{-1}$, and then the samples were heated at 180 °C for 10 min in glovebox, achieving PV component in an architecture of glass/FTO/ETL(c/m-TiO$_2$)/perovskite/HTL (Spiro-PT).

**Monolithic PVCD fabrication**. The as-cleaned FTO/glass substrates (2 cm × 2.5 cm) were treated by UV-ozone for 30 min. Then 300 μL EC gel was cast uni-formly on the FTO side and kept drying in air for 12 h. The resulting EC gel was cut into an active area of 2 cm × 2 cm. The EC gel component was pasted onto the top of PV component, forming glass/FTO/PV/EC gel/FTO/glass configuration of the device. Subsequently, side faces of both EC and PV submodules were fixed firmly using UV curing adhesive. Finally, two FTO electrodes in PVCD were connected with thin copper wire and conductive copper glue, inducing an external electron-transporting channel between two FTO electrodes.

**In situ T% measurement**. The in situ T% spectra were recorded by using AM1.5G solar simulator (San-EI Electric, XES-40S3) as the illuminant via an optical fiber with a spectrometer (OceanOptics, Maya 2000, VIS-NIR) at the 350–1000 nm range.

**Long-term repeatability test**. For repeatability test, the T% spectra were recorded by using a halogen lamp (OceanOptics, HL-2000) as the illuminant and a 395 nm LED (10 Watt) with an electronic shutter (Daheng Optics, GCI-73M) as the control light (Supplementary Fig. 9). In each cycle, PVCD was illuminated for 10 s under UV light for coloring process and blocked by shutter for 45 s for bleaching process, periodically. The dynamic T% intensity at 600 nm were collected by spectrometer (OceanOptics, Maya 2000, VIS-NIR). After recording the initial 100 cycles, the two-terminal FTO electrode was short connected by using a copper wire for continuously operation under UV irradiation for 1-week (ca. 11,000 cycles) in glovebox. Afterwards, the final 100 cycles were collected again by spectrometer in dynamic model.

**Time-of-flight secondary-ion mass spectrometry (ToF-SIMS)**. A TOF-SIMS was used to depth-profile the perovskite materials and completed devices. Analysis was completed using a 3-lens 30-kV Bi-Mn primary ion gun, the Bi$^+$ primary-ion beam (operated in bunched mode; 10-ns pulse width, analysis current of 1.0 pA) was scanned over a 25 × 25 μm area. Depth profiling was accomplished with a 1 kV oxygen-ion sputter beam (10.8 nA sputter current) raster of 80 × 80 μm area. All spectra during profiling were collected at a primary ion dose density of $1 \times 10^{12}$ ions cm$^{-2}$ to remain at the static-SIMS limit.

**Photoelectron yield spectroscopy (PYS)**. PYS measurements were conducted using on an AC-2 photoelectron spectrometer (Riken-Keiki Co.).

**Scanning electron microscopy (SEM)**. The top-view and side-view morphologies were investigated by using field-emission scanning electron microscope (JEOL, JSM-7800F).

**Haze characterizations**. The haze of perovskite films were measured by a haze-meter (Diffusion Systems UK, EEL 57D)

**Solar cell characterizations**. The photocurrent density-voltage curves of the perovskite solar cells were measured using a solar simulator (Oriel 94023A, 300 W) and a Keithley 2400 source meter. The intensity (100 mW cm$^{-2}$) was calibrated using a standard Si solar cell (Oriel, VLSI standards). All the devices were tested under AM 1.5G sun light (100 mW cm$^{-2}$) using a metal mask of 0.1 cm$^2$ with a scan rate of 10 mV s$^{-1}$. IPCE measurements were carried out in DC mode by using a Keithley 2400 source meter and a SOFN 7ISW752 monochromator equipped with a 500 W Xenon lamp. The wavelength sampling interval was 5 nm and the current sampling time was 1 s, which was fully controlled by computer. A Hamamatsu S1337-1010BQ silicon diode used for IPCE measurements was cali-brated at the National Institute of Metrology, China.

## Data availability
All relevant data that support the finding of this study are available from the corresponding author upon reasonable request.

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

## Acknowledgements

This work is supported financially by the National Key Research and Development Program of China (2017YFE0131900); National Natural Science Foundation of China (91833306, 62075094, 52003118); the Recruitment Program of Global Experts; the Innovation and Entrepreneurship Program of Jiangsu Province; the Six Talent Peak Project of Jiangsu Province. We thank K. Wen from Nanjing Tech University for programming the software of in situ $V_{OC}$ and $I_{SC}$ measurement. We thank Cho et al., *Semitransparent Energy-Storing Functional Photovoltaics Monolithically Integrated with Electrochromic Supercapacitors* of *Advanced Functional Materials*, Wiley Publisher (Copyright License Nr. 5053091442213) for inspiring Fig. 4a.

## Author contributions

T.Q. conceived the idea and designed the experiments. T.Q. and W.H. supervised the work. Y.L. carried out PV element fabrication and characterizations. Jungan W. carried out EC element fabrication and characterizations. F.W. carried out ToF-SIMS measurements and commented manuscript revision. Z.C. carried out haze rate, Y.F. carried out SEM, Q.C. carried out electrochemical, and J.Z. carried out GIWAXS measurements. Y.L. and Jungan W. conducted the PVCD integration and characterizations, with the assistance of L.W. and Jianpu W. proposed the PVCD test methods. T.Q. wrote the first draft of the manuscript and W.H. provided major revisions. All authors discussed the results and commented on the manuscript.

## Competing interests

The authors declare no competing interests.
