## [Peer Review File · Nature Communications]

REVIEWER COMMENTS

Reviewer #1 (Remarks to the Author):

This study by Liu et al is interesting and the data are excellent. Authors have some new findings, which may cause broad interests in the related research fields. Authors used a few technologies to demonstrate their hypothesis for their new findings, however, some explanations are difficult to understand or obscure. I suggest a major revision before acceptance.

Abstract: authors stated: "We believe this initial Turing-patterned photoelectronic prototype can reveal a new bionic strategy to break through the limit of Moore's law against traditional photolithographic technology". I agree with authors the data are good, however, this description is exaggerated too much, I suggest authors reword.

Also authors stated that the pattern is a kind of Turing Patterns. I have different opinions. I just saw some irregular patterns with cracks. I don't think the emphasis of Turing pattern is very necessary. If possible, please consider to reword.

Authors said they demonstrated an outstanding >10000 cycle reproducibility and one-week service period (Fig. 3f). However, I saw significant transmittance change even only in the last 100 cycles of Figure 3f. I think the decay percentage is needed, not just show the reproducibility can be over 10000. I suggest authors add a discussion about the stability comparison with others' state-of-art work for the PVCD-based smart windows.

Authors said: "Unfortunately, until now, all previous photovoltachromic device (PVCD) prototypes had diverse drawbacks, such as liquid electrolyte leakage, non-full frame, low transmittance, complex circuits, etc." This sentence is too absolute, I suggest authors to revise. Although authors gave a comparison Extended Table 2, this table does not have device performance like stability, reversibility, etc.

I suggest authors remove "We thus believe this work is the most evolutionary solar-driven smart window technology until now."

Authors did some experiments to give reasons about the high transmittance of the film and attribute the high transmittance to inclined light refractions on vertical interfaces of Turing-patterned voids. Why does the light need or have refractions here?

There are many small square in the supplementary information, for example: at 60 SQUARE SYMBOL for 30 min; 180 SQUARE SYMBOL for 10 min. There are many. These squares might be generated during the file transformation from word to PDF. Please change accordingly.

Extended Data Table 1. Does the comparison about the transmittance need consideration of thickness?

Extended Data Fig. 8 | "Photograhs" should be "photographs".

Reviewer #2 (Remarks to the Author):

Liu et al fabricate monolithic (two-terminal) chromophotovoltaic devices using a transparent metal halide perovskite material. The transparent perovskite is formed by reacting MACl with a MAPbBr₃ film. The MACl reaction proceeds through a reaction-diffusion reaction, which yield nanoscale patterning of the absorber with a bandgap that tunes absorption outside of the visible spectrum. They have nice results that get wrapped up in what seems to be contrived storytelling with an emphasis on "Turing reactions." It is not clear that the voids improve performance in optical properties or photovoltachromic device performance. Though the story is compelling, they show no comparison to films of the same stoichiometry without voids. The work is incomplete without this analysis, and it is clear from figure 1 that they are capable of making such films and devices. They need to answer the simple question: Is the patterning needed? The authors must answer this

question before the paper can be considered for publication. If it is not needed, then the results are still compelling, but the Turing narrative should be removed. Here are my more specific comments:

The effect on bandgap is clearly important and likely more important than the Turing patterns for fabricating a transparent device. Extended Data Fig. 5 is critical to the story and should be emphasized more than the Turing reaction narrative.

The authors are clearly capable of complex optical measurements. I have a few issues with the analysis:

1) An attribute they do not recognize but does carry weight for window technologies is haze. The scale of their patterned absorber should result in less haze than strategies that form microscope patterns. For instance, the authors would be correct to claim improvement over the Snaith paper (ref. 72), which uses microscopic patterns. The work here should have low haze and will fit nicely into their discussion about transmittance as a function of angle. The authors should evaluate the haze of the devices as an important metric for their patterned strategy and should be compared against unpatterned films with the same composition/bandgap.

2) The authors claim the voids improve transmissivity, but they don't have the data to back this up. Fig. 2 is dedicated to transmissivity measurements, and they show AVT increases with longer MACL soak times. They then claim the voids improve transmissivity. However, the change in bandgap convolutes this point. The authors should repeat these measurements on MAPbBrxCl_{3-x} films that do not have voids. It is clear from Fig. 1a that they are capable fabricating such films. They do mention the "homogenous counterpart," but I do not see the data in the paper or supporting information.

3) The authors discuss color of the PVCD, but do not evaluate color of the perovskite film. The authors state: "In the Photographs of PVCD under 0.1 sun illumination clearly showed an excellent colour rendering index (CRI) of 96 (Fig. 3c)", but the film clearly appear yellow in the bleached state and blue in the colored state. The authors should better discuss the color neutrality of their films.

The simplicity, cycle stability, and optical performance of the PVCD devices is impressive, but, again, it is not clear what role the voids play. Devices with and without voids should be evaluated.

Smaller issues I noticed:

Fig 2d has ϕ_r as a x-axis label. This is not defined.

Whereas claims about haze would be correct, the authors make inappropriate comparisons to existing literature for patterned perovskite films for semitransparent PV. "This extremely high AVT is considerably ahead of all previous literatures using microscale patterning²³⁻²⁵ 70 semi71 transparent PSKs (Extended Data Table. 1). In these works, PSK films were fabricated in island⁷² or network morphologies, which can only pass visible light in partial void areas and thus lead to 73 poor transmittance to maximum 50% of AVT." They are comparing to semitransparent devices designed for a balance between high power conversion and visible transmittance. Their power conversion is significantly higher than the work shown here, and this can only be achieved if the visible part of the solar spectrum is absorbed. The tradeoff is summarized here:
10.1021/acseenergylett.9b01316

The works of Lunt (e.g. 10.1038/s41560-017-0016-9) and Loo (10.1038/nenergy.2017.104) are clearly missing from the narrative. Thermochromic/switchable PV windows based on perovskites as the active switching material has also been ignored but clearly a relevant technology. These are more important works to highlight than the narrative of Turing reactions.

Reviewer #3 (Remarks to the Author):

The manuscript "Turing Reaction-Diffusion Strategy for Perovskite Photovoltachromic Smart Windows" by Liu et al presents a study about smart window applications with the use of perovskite solar cell. The authors perform Turing's reaction diffusion model of mixed-halide perovskites. They further construct perovskite PV cell and connect it with electrochromic component. This hybrid device exhibit chromic properties, which can be used for smart window applications.

In terms of the novelty of this design, the perovskite solar cell powered electrochromic device for smart window applications have been reported in a number of existing literatures, including Mater. Horiz., 2016, 3, 588-595; Chem. Commun., 2019,55, 12060-12063. This work didn't show substantial advances compared with existing works.

Regarding Turing's reaction diffusion model, the authors present the morphology characterization results, but didn't present fundamental insight.

In the abstract, the authors claim "new bionic strategy to break through the limit of Moore's law against traditional photolithographic technology". Moore's law is about the downward scaling of Si transistor. What's the relationship to this work?

Considering the novelty, insight, presentation and coherence of this manuscript, I feel that this work is still premature for the publication in high-impact journals, like Nature Communications.

Reply to reviewer #1 (Remarks to the Author)

This study by Liu et al is interesting and the data are excellent. Authors have some new findings, which may cause broad interests in the related research fields. Authors used a few technologies to demonstrate their hypothesis for their new findings, however, some explanations are difficult to understand or obscure. I suggest a major revision before acceptance.

Response: We thank the reviewer very much for your insightful comments. We have revised our manuscript accordingly as detailed below.

Comment #1: Abstract: authors stated: “We believe this initial Turing-patterned photoelectronic prototype can reveal a new bionic strategy to break through the limit of Moore's law against traditional photolithographic technology”. I agree with authors the data are good, however, this description is exaggerated too much, I suggest authors reword.

Response: We thank the referee point out this question. We have deleted the exaggerated content mentioned by the reviewers in the revised manuscript. (Page 1, line 16-39)

Comment #2: Also authors stated that the pattern is a kind of Turing Patterns. I have different opinions. I just saw some irregular patterns with cracks. I don't think the emphasis of Turing pattern is very necessary. If possible, please consider to reword.

Response: We thank the reviewer for pointing out this important point. We agree with the reviewer's opinion, and have removed any content related to the Turing pattern and rewrote our manuscript.

Comment #3: Authors said they demonstrated an outstanding >10000 cycle reproducibility and one-week service period (Fig. 3f). However, I saw significant transmittance change even only in the last 100 cycles of Figure 3f. I think the decay percentage is needed, not just show the reproducibility can be over 10000. I suggest authors add a discussion about the stability comparison with others' state-of-art work for the PVCD-based smart windows.

Response: Thanks for the reviewer's suggestion. According to the reviewer's suggestion, we have redrawn Fig. 3f, and added discussion on device stability compared with previous work and marked it in yellow. (Page 8, line 215-2167)

Comment #4: Authors said: “Unfortunately, until now, all previous photovoltachromic device (PVCD) prototypes had diverse drawbacks, such as liquid electrolyte leakage, non-full frame, low transmittance, complex circuits, etc.” This sentence is too absolute, I suggest authors to revise. Although authors gave a comparison Extended Table 2, this table does not have device

performance like stability, reversibility, etc.

Response: Thanks for the reviewer's suggestion. We have reworded the relevant contents mentioned by the reviewers in the revised manuscript and added the stability or reversibility of PVCD in revised Supplementary Table 2. (Page 5, Fig. 140-142)

Comment #5: I suggest authors remove “We thus believe this work is the most evolutionary solar-driven smart window technology until now.”

Response: We thank the reviewer for pointing out this important point. We have deleted this sentence according to the reviewer's suggestion.

Comment #6: Authors did some experiments to give reasons about the high transmittance of the film and attribute the high transmittance to inclined light refractions on vertical interfaces of Turing-patterned voids. Why does the light need or have refractions here?

Response: We thank the reviewer for pointing out this important point. We admit that Turing pattern is not needed according to the reviewer's suggestion. For the rigor of manuscript narration, we have removed the relevant content about Turing narrative and its corresponding characterization tests.

Comment #7: There are many small square in the supplementary information, for example: at 60 SQUARE SYMBOL for 30 min; 180 SQUARE SYMBOL for 10 min. There are many. These squares might be generated during the file transformation from word to PDF. Please change accordingly.

Response: We thank the reviewer for pointing out this point. This is our mistake, we have corrected it in the supplementary information and marked it with yellow. (Supplementary information Page 3, line 51-77 and Page 5, line 117)

Comment #8: Extended Data Table 1. Does the comparison about the transmittance need consideration of thickness?

Response: We thank the reviewer for pointing out this important topic. We admit that the transmittance was related to the thickness of the film. For PVCD, researchers usually try to make the PV films as thin as possible on the premise that photovoltaic could drive the electrochromic device to work, for achieving the maximum transmittance of PVCD approaching the actual natural window. Therefore, we believed that the comparison in Supplementary Table 1 was reasonable. However, for the sake of narrative rigor, we revised the comparison between our work and previous research and revised Supplementary Table 1. (Page 2, line 62-64)

Comment #9: Extended Data Fig. 8 | “Photograh” should be “photographs”.

Response: We thank the reviewer for pointing out this important point. We have corrected it in the supplementary materials and marked it with yellow. (Supplementary materials Page 7, line 158)

Reply to reviewer #2 (Remarks to the Author)

Liu et al fabricate monolithic (two-terminal) chromophotovoltaic devices using a transparent metal halide perovskite material. The transparent perovskite is formed by reacting MAI with a MAPbBr₃ film. The MAI reaction proceeds through a reaction-diffusion reaction, which yield nanoscale patterning of the absorber with a bandgap that tunes absorption outside of the visible spectrum. They have nice results that get wrapped up in what seems to be contrived storytelling with an emphasis on “Turing reactions.” It is not clear that the voids improve performance in optical properties or photovoltaic device performance. Though the story is compelling, they show no comparison to films of the same stoichiometry without voids. The work is incomplete without this analysis, and it is clear from figure 1 that they are capable of making such films and devices. They need to answer the simple question: Is the patterning needed? The authors must answer this question before the paper can be considered for publication. If it is not needed, then the results are still compelling, but the Turing narrative should be removed.

Response: We thank the reviewer very much for your insightful comments. After careful consideration, we admit that Turing pattern is not needed, and according to the reviewer's suggestion, we have removed the relevant content about Turing narrative. We have extensively revised the manuscript accordingly as detailed below.

Comment #1: The effect on bandgap is clearly important and likely more important than the Turing patterns for fabricating a transparent device. Extended Data Fig. 5 is critical to the story and should be emphasized more than the Turing reaction narrative.

Response: We thank the referee point out this question. According to your suggestion, we have deleted the contents about the Turing reaction narrative, and a detailed discussion of Fig. 1c have been added in the manuscript. (We marked it with yellow on Page 3, line 91-94)

Comment #2: An attribute they do not recognize but does carry weight for window technologies is haze. The scale of their patterned absorber should result in less haze than strategies that form microscope patterns. For instance, the authors would be correct to claim improvement over the Snaith paper (ref. 72), which uses microscopic patterns. The work here should have low haze and will fit nicely into their discussion about transmittance as a function of angle. The authors should evaluate the haze of the devices as an important metric for their patterned strategy and should be compared against unpatterned films with the same composition/bandgap.

Response: We thank the reviewer for pointing out this important topic. We have tested the haze

value of the film and added a discussion on the important performance parameters of haze in the revised manuscript. (We marked it with yellow on Page 3-4, line 96-100)

Comment #3: The authors claim the voids improve transmissivity, but they don't have the data to back this up. Fig. 2 is dedicated to transmissivity measurements, and they show AVT increases with longer MACl soak times. They then claim the voids improve transmissivity. However, the change in bandage convolutes this point. The authors should repeat these measurements on MAPbBr_xCl_{3-x} films that do not have voids. It is clear from Fig. 1a that they are capable fabricating such films. They do mention the "homogenous counterpart," but I do not see the data in the paper or supporting information.

Response: We thank the referee point out this question. We agree with the reviewers that Turing patterns are not essential, so we have deleted the content text about the voids according the reviewer's suggestion.

Comment #4: The authors discuss color of the PVCD, but do not evaluate color of the perovskite film. The authors state: "In the Photographs of PVCD under 0.1 sun illumination clearly showed an excellent colour rendering index (CRI) of 96 (Fig. 3c)", but the film clearly appear yellow in the bleached state and blue in the colored state. The authors should better discuss the color neutrality of their films.

Response: Thanks for the reviewer's suggestion. We do not fully agree with the reviewer's suggestion. First, the color rendering index value of 96 is calculated according to the in-situ transmittance characterization test. Secondly, the fading PVCD looks yellow, which may be due to the contrast correction when shooting by the camera. In fact, it looks much lighter (light yellow of thin EC gels). (Supplementary Fig. 8c, Page 9)

Comment #5: The simplicity, cycle stability, and optical performance of the PVCD devices is impressive, but, again, it is not clear what role the voids play. Devices with and without voids should be evaluated.

Response: Thanks for the reviewer's suggestion. Firstly, we have deleted the content text about the Turing pattern. Secondly, we admit that the gap only plays a minor role in improving the transmittance of films. On the contrary, the band gap of the film is the main factor affecting the transmittance. Thirdly, due to the fast crystallization rate of perovskite precursor solution, it is not possible for us to prepare high quality films with the same stoichiometry without voids to compare with those with voids (in Fig. 1b). (Page 3)

Comment #6: Fig 2d has phi_r as a x-axis label. This is not defined.

Response: We thank the referee point out this question. We have removed the content of Turing pattern in the revised manuscript.

Comment #7: Whereas claims about haze would be correct, the authors make inappropriate comparisons to existing literature for patterned perovskite films for semitransparent PV. “This extremely high AVT is considerably ahead of all previous literatures using microscale patterning semi-transparent PSKs (Extended Data Table. 1). In these works, PSK films were fabricated in island or network morphologies, which can only pass visible light in partial void areas and thus lead to poor transmittance to maximum 50% of AVT.” They are comparing to semitransparent devices designed for a balance between high power conversion and visible transmittance. Their power conversion is significantly higher than the work shown here, and this can only be achieved if the visible part of the solar spectrum is absorbed. The tradeoff is summarized here: [10.1021/acseenergylett.9b01316](https://doi.org/10.1021/acseenergylett.9b01316)

Response: We thank the referee point out this question and provide thus useful reference. We accept the advice on haze and add this experiment data in line 96-100, page 3-4. In addition, we agree that a balance between AVT and PCE is the tradeoff for semi-transparent PVs, which are more focusing on the power-harvesting properties in BIPV applications. Nevertheless, in this work we putted more attention on smarter windows needing more critical requirements on transmittance for indoor illumination and colour naturality. Therefore, we consider the transmittance is more important in this work, since the UV-harvesting PV layer already provided enough power to drive smart window system.

Comment #8: The works of Lunt (e.g. [10.1038/s41560-017-0016-9](https://doi.org/10.1038/s41560-017-0016-9)) and Loo ([10.1038/nenergy.2017.104](https://doi.org/10.1038/nenergy.2017.104)) are clearly missing from the narrative. Thermochromic/switchable PV windows based on perovskites as the active switching material has also been ignored but clearly a relevant technology. These are more important works to highlight than the narrative of Turing reactions.

Response: We thank the referee point out this question. We have added the works of Lunt (e.g. [10.1038/s41560-017-0016-9](https://doi.org/10.1038/s41560-017-0016-9)) and Loo ([10.1038/nenergy.2017.104](https://doi.org/10.1038/nenergy.2017.104)) on page 2, line 74-76, and page 2, line 47, respectively. In the abstract of the revised manuscript, we narrate the thermochromic or switchable photovoltaic window based on perovskite.

Reply to reviewer #3 (Remarks to the Author)

The manuscript “Turing Reaction-Diffusion Strategy for Perovskite Photovoltachromic Smart Windows” by Liu et al presents a study about smart window applications with the use of perovskite solar cell. The authors perform Turing’s reaction diffusion model of mixed-halide perovskites. They further construct perovskite PV cell and connect it with electrochromic component. This hybrid device exhibit chromic properties, which can be used for smart window applications.

Response: We thank the reviewer very much for your insightful comments. We have revised our manuscript accordingly as detailed below.

Comment #1: In terms of the novelty of this design, the perovskite solar cell powered electrochromic device for smart window applications have been reported in a number of existing literatures, including Mater. Horiz., 2016, 3, 588-595; Chem. Commun., 2019,55, 12060-12063. This work didn’t show substantial advances compared with existing works.

Response: We thank the referee point out this question. We admitted that previous studies have reported electrochromic devices powered by perovskite solar cells for smart window applications, but our work was different from previous work in three main points: 1. Our PVCD is full-frame; 2. Our PVCD is a monolithic structure with two endpoints; 3. The AVT of our PVCD exceeds 75% is superior to previous work. 4. Our PVCD can be controlled on demand and self-adapted based on solar radiation.

Comment #2: Regarding Turing’s reaction diffusion model, the authors present the morphology characterization results, but didn’t present fundamental insight.

Response: We thank the referee point out this question. In order to make the manuscript more rigorous and coherent. We have deleted the Turing’s reaction diffusion model in the manuscript.

Comment #3: In the abstract, the authors claim “new bionic strategy to break through the limit of Moore's law against traditional photolithographic technology”. Moore’s law is about the downward scaling of Si transistor. What’s the relationship to this work?

Response: We thank the referee point out this question. According to the reviewer's suggestion, we deleted the relevant content.

Comment #4: Considering the novelty, insight, presentation and coherence of this manuscript, I feel that this work is still premature for the publication in high-impact journals, like Nature Communications.

Response: We thank the referee point out this question. According to the reviewer's opinion, we carefully revised the unreasonable part. We sincerely request the reviewer to reconsider our manuscript carefully.

REVIEWERS' COMMENTS

Reviewer #1 (Remarks to the Author):

Authors addressed my questions and improved the manuscript. Since the previous confusing part about "Turing Pattern" was removed by the authors, I do not have further questions. I think the manuscript can be accepted at current format.

Reviewer #2 (Remarks to the Author):

The authors have addressed my main concerns with the "Turing pattern" by removing it. They have also addressed my other comments, and I think manuscript has come back with a stronger focus on the important points of the work. I recommend publication and congratulate the authors on their impactful research.

Reviewer #3 (Remarks to the Author):

The authors revised claims by deleting some sentences. But the questions still remain there. This version is still premature for publication in a high-impact journal like Nature Communications.

Point-by-Point Response to Referees

Reply to reviewer #1 (Remarks to the Author)

Comment #1: Authors addressed my questions and improved the manuscript. Since the previous confusing part about “Turing Pattern” was removed by the authors, I do not have further questions. I think the manuscript can be accepted at current format.

Response: We thank the reviewer for their support of our manuscript.

Reply to reviewer #2 (Remarks to the Author)

Comment #1: The authors have addressed my main concerns with the "Turing pattern" by removing it. They have also addressed my other comments, and I think manuscript has come back with a stronger focus on the important points of the work. I recommend publication and congratulate the authors on their impactful research.

Response: We thank the reviewer for agreeing to accept.

Reply to reviewer #3 (Remarks to the Author)

Comment #1: The authors revised claims by deleting some sentences. But the questions still remain there. This version is still premature for publication in a high-impact journal like Nature Communications.

Response: We still thank the reviewer for their excellent suggestions on this work during the review process.